# Variational Information Bottleneck for Unsupervised Clustering: Deep Gaussian Mixture Embedding

## Abstract

In this paper, we develop an unsupervised generative clustering framework that combines variational information bottleneck and the Gaussian Mixture Model. Specifically, in our approach we use the variational information bottleneck method and model the latent space as a mixture of Gaussians. We derive a bound on the cost function of our model that generalizes the evidence lower bound (ELBO); and provide a variational inference type algorithm that allows to compute it. In the algorithm, the coders' mappings are parametrized using neural networks and the bound is approximated by Markov sampling and optimized with stochastic gradient descent. Numerical results on real datasets are provided to support the efficiency of our method.

## 1 Introduction

Clustering consists in partitioning a given data set into various groups (clusters) based on some similarity metric, such as Euclidean distance, $L_1$ norm, $L_2$ norm, $L_\infty$ norm, the popular logarithmic loss measure or others. The principle is that each cluster should contain elements of the data that are closer to each other than to any other element outside that cluster, in the sense of the defined similarity measure. If the joint distribution of the clusters and data is not known, one should operate blindly in doing so, i.e., using only the data elements at hand; and the approach is called unsupervised clustering. Unsupervised clustering is perhaps one of the most important tasks of unsupervised machine learning algorithms nowadays, due to a variety of application needs and connections with other problems.

Examples of unsupervised clustering algorithms include the so-popular $K$-means (Hartigan & Wong, 1979) and expectation maximization (EM) (Dempster et al., 1977). The $K$-means algorithm partitions the data in a manner that the Euclidean distance among the members of each cluster is minimized. With the EM algorithm, the underlying assumption is that the data comprises a mixture of Gaussian samples, namely a Gaussian Mixture Model (GMM); and one estimates the parameters of each component of the GMM while simultaneously associating each data sample to one of those components. Although they offer some advantages in the context of clustering, these algorithms suffer from some strong limitations. For example, it is well known that the $K$-means is highly sensitive to both the order of the data and scaling; and the obtained accuracy depends strongly on the initial seeds (in addition to that it does not predict the number of clusters or $K$-value). The EM algorithm suffers mainly from low convergence, especially for high dimensional data.

Recently, a new approach has emerged that seeks to perform inference on a transformed domain (generally referred to as latent space), not the data itself. The rationale is that because the latent space often has fewer dimensions it is more convenient computationally to perform inference (clustering) on it rather than on the high dimensional data directly. A key aspect then is how to design a latent space that is amenable to accurate low-complex unsupervised clustering, i.e., one that preserves only those features of the observed high dimensional data that are useful for clustering while removing out all redundant or non-relevant information. Along this line of work, we can mention (Ding & He, 2004) which utilizes Principal Component Analysis (PCA) (Wold et al., 1987) for dimensionality reduction followed by $K$-means for clustering the obtained reduced dimension data; or (Roweis, 1997) which uses a combination of PCA and the EM algorithm. Other works that use alternatives

for the linear PCA include Kernel PCA (Hofmann et al., 2008), which employs PCA in a non-linear fashion to maximize variance in the data.

The usage of deep neural networks (DNN) for unsupervised clustering of high dimensional data on a lower dimensional latent space has attracted considerable attention, especially with the advent of autoencoder (AE) learning and the development of powerful tools to train them using standard backpropagation techniques (Kingma & Welling, 2014; Rezende et al., 2014). Advanced forms include Variational autoencoders (VAE) (Kingma & Welling, 2014; Rezende et al., 2014) which are generative variants of AE that regularize the structure of the latent space and the more general Variational Information Bottleneck (VIB) of (Alemi et al., 2017) which is a technique that is based on the Information Bottleneck method (Tishby et al., 1999) and seeks a better trade-off between accuracy and regularization than VAE via the introduction of a Lagrange-type parameter $s$ which controls that trade-off and whose optimization is similar to deterministic annealing (Slonim, 2002) or stochastic relaxation.

In this paper, we develop an unsupervised generative clustering framework that combines VIB and the Gaussian Mixture Model. Specifically, in our approach we use the variational information bottleneck method and model the latent space as a mixture of Gaussians. The encoder and decoder of the model are parametrized using neural networks (NN). The cost-function is calculated approximatively by Markov sampling and optimized with stochastic gradient descent. Furthermore, the application of our algorithm to the unsupervised clustering of various datasets, including the MNIST (Lecun et al., 1998), REUTERS (Lewis et al., 2004) and STL-10 (Coates et al., 2011), allows a better clustering accuracy than previous state of the art algorithms. For instance, we show that our algorithm performs better than the variational deep embedding (VaDE) algorithm of (Jiang et al., 2017) which is based on VAE and performs clustering by maximizes the ELBO and can be seen as a specific case of our algorithm (Section 3.1). Our algorithm also generalizes the VIB of (Alemi et al., 2017) which models the latent space as an isotropic Gaussian which is generally not expressive enough for the purpose of unsupervised clustering. Other related works, but which are of lesser relevance to the contribution of this paper, are the deep embedded clustering (DEC) of (Xie et al., 2016), the improved deep embedded clustering (IDEC) of (Guo et al., 2017) and (Dilokthanakul et al., 2017). For a detailed survey of clustering with deep learning, the readers may refer to (Min et al., 2018).

To the best of our knowledge, our algorithm performs the best in terms of clustering accuracy by using deep neural networks without any prior knowledge regarding the labels (except the usual assumption regarding the number of the classes) compared to the state-of-the-art algorithms of this category. In order to achieve the aforementioned accuracy, i) we derive a cost-function that contains the IB hyper parameter $s$ that controls the trade-off between over-fit and generalization of the model and we used an approximation of KL divergence that avoid assumptions which do not hold in the beginning of the learning process and lead to convergence issues; ii) evaluate the hyper-parameter $s$ by following an annealing approach that improves both the convergence and the accuracy of the proposed algorithm.

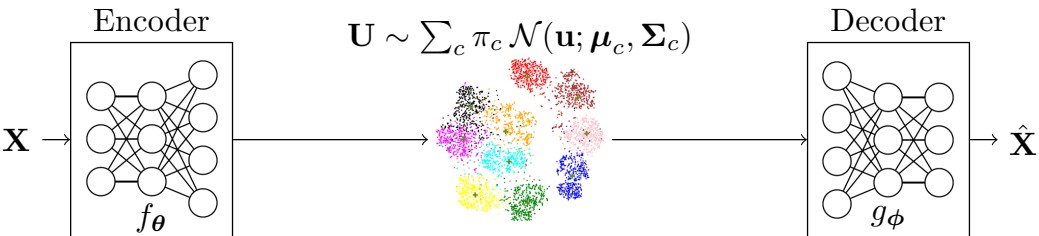

Figure 1: Variational Information Bottleneck with Gaussian Mixtures.

## 2 PROBLEM DEFINITION AND MODEL

Consider a dataset that is composed of $N$ samples $\{\mathbf{x}_i\}_{i=1}^N$ which we wish to partition into $|\mathcal{C}| \geq 1$ clusters. Let $\mathcal{C} = \{1, \ldots, |\mathcal{C}|\}$ be the set of all possible clusters; and $C$ designate a categorical random variable that lies in $\mathcal{C}$ and stands for the index of the actual cluster. If $\mathbf{X}$ is a random variable that models elements of the dataset, given $\mathbf{X} = \mathbf{x}_i$ induces a probability distribution on $C$ which the learner should learn. Thus, mathematically the problem is that of estimating the values of the

unknown conditional probability $P_{C|\mathbf{X}}(\cdot|\mathbf{x}_i)$ for all elements $\mathbf{x}_i$ of the dataset. The estimates are sometimes referred to as the assignment probabilities.

As mentioned previously, we use the VIB framework and model the latent space as a GMM. The resulting model is depicted in Figure 1, where the parameters $\pi_c$, $\boldsymbol{\mu}_c$, $\boldsymbol{\Sigma}_c$, for all values of $c \in \mathcal{C}$, are to be optimized jointly with those of the employed NNs as instantiation of the coders. Also, the assignment probabilities are estimated based on the values of latent space vector instead of the observation themselves, i.e., $P_{C|\mathbf{U}} = Q_{C|\mathbf{U}}$. In the rest of this section, we elaborate on the inference and generative network models for our method, which are illustrated below.

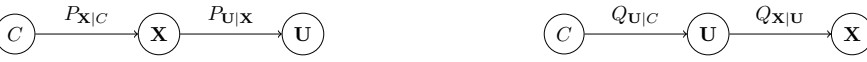



Figure 2: Inference Network        Figure 3: Generative Network



## 2.1 INFERENCE NETWORK MODEL

We assume that an observed data $\mathbf{x}$ is generated from a GMM with $|\mathcal{C}|$ components. Then, the latent representation $\mathbf{u}$ is inferred according the following procedure:

1. One of the components of the GMM is chosen according to a categorical variable $C$.

2. The data $\mathbf{x}$ is generated from the $c$-th competent of the GMM, i.e., $P_{\mathbf{X}|C} \sim \mathcal{N}(\mathbf{x}; \tilde{\boldsymbol{\mu}}_c, \tilde{\boldsymbol{\Sigma}}_c)$.

3. Encoder maps $\mathbf{x}$ to a latent representation $\mathbf{u}$ according to $P_{\mathbf{U}|\mathbf{X}} \sim \mathcal{N}(\boldsymbol{\mu}_\theta, \boldsymbol{\Sigma}_\theta)$.

   3.1. The encoder is modeled with a DNN $f_\theta$ which maps $\mathbf{x}$ to the parameters of a Gaussian distribution, i.e., $[\boldsymbol{\mu}_\theta, \boldsymbol{\Sigma}_\theta] = f_\theta(\mathbf{x})$.

   3.2. The representation $\mathbf{u}$ is sampled from $\mathcal{N}(\boldsymbol{\mu}_\theta, \boldsymbol{\Sigma}_\theta)$.

For the inference network, shown in Figure 2, the following Markov chain holds

$$C \,-\!\circ\!-\, \mathbf{X} \,-\!\circ\!-\, \mathbf{U} \ . \tag{1}$$

## 2.2 GENERATIVE NETWORK MODEL

Since encoder extracts useful representations of the dataset and we assume that the dataset is generated from a GMM, we model our latent space also with a mixture of Gaussians. To do so, the categorical variable $C$ is embedded with the latent variable $\mathbf{U}$. The reconstruction of the dataset is generated according to the following procedure:

1. One of the components of the GMM is chosen according to a categorical variable $C$, with a prior distribution $Q_C$.

2. The representation $\mathbf{u}$ is generated from the $c$-th component, i.e., $Q_{\mathbf{U}|C} \sim \mathcal{N}(\mathbf{u}; \boldsymbol{\mu}_c, \boldsymbol{\Sigma}_c)$.

3. The decoder maps the latent representation $\mathbf{u}$ to $\hat{\mathbf{x}}$ which is the reconstruction of the source $\mathbf{x}$ by using the mapping $Q_{\mathbf{X}|\mathbf{U}}$.

   3.1. The decoder is modeled with a DNN $g_\phi$, that maps $\mathbf{u}$ to the estimate $\hat{\mathbf{x}}$, i.e., $[\hat{\mathbf{x}}] = g_\phi(\mathbf{u})$.

For the generative network, shown in Figure 3, the following Markov chain holds

$$C \,-\!\circ\!-\, \mathbf{U} \,-\!\circ\!-\, \mathbf{X} \ . \tag{2}$$

## 3 PROPOSED METHOD

In this section we present our clustering method. First, we provide a general cost function for the problem of the unsupervised clustering that we study here based on the variational IB framework; and we show that it generalizes the ELBO bound developed in (Jiang et al., 2017). We then parametrize our model using NNs whose parameters are optimized jointly with those of the GMM. Furthermore, we discuss the influence of the hyper-parameter $s$ that controls optimal trade-offs between accuracy and regularization.

### 3.1 Brief review of variational Information Bottleneck for unsupervised learning

As described in Chapter 2, the stochastic encoder $P_{\mathbf{U}|\mathbf{X}}$ maps the observed data $\mathbf{x}$ to a representation $\mathbf{u}$. Similarly, the stochastic decoder $Q_{\mathbf{X}|\mathbf{U}}$ assigns an estimate $\hat{\mathbf{x}}$ of $\mathbf{x}$ based on the vector $\mathbf{u}$. As per the IB method (Tishby et al., 1999) a suitable representation $\mathbf{U}$ should strike the right balance between capturing all information about the categorical variable $C$ that is contained in the observation $\mathbf{X}$ and using the most concise representation for it. This leads to maximizing the following Lagrange problem

$$\mathcal{L}_s(\mathbf{P}) = I(C; \mathbf{U}) - sI(\mathbf{X}; \mathbf{U}) , \tag{3}$$

where $s \geq 0$ designates the Lagrange multiplier and for convenience $\mathbf{P}$ denotes the conditional distribution $P_{\mathbf{U}|\mathbf{X}}$.

Instead of equation 3 which is not always computable in our unsupervised clustering setting, we use a modified version of it (so-called unsupervised IB objective (Alemi et al., 2017)) given by

$$\tilde{\mathcal{L}}_s(\mathbf{P}) := -H(\mathbf{X}|\mathbf{U}) - s[H(\mathbf{U}) - H(\mathbf{U}|\mathbf{X})] \tag{4}$$

$$= \mathbb{E}_{P_{\mathbf{X}}} \left[ \mathbb{E}_{P_{\mathbf{U}|\mathbf{X}}}[\log P_{\mathbf{X}|\mathbf{U}} + s \log P_{\mathbf{U}} - s \log P_{\mathbf{U}|\mathbf{X}}] \right] . \tag{5}$$

For a variational distribution $Q_{\mathbf{U}}$ on $\mathcal{U}$ (instead of the unknown $P_{\mathbf{U}}$) and a variational stochastic decoder $Q_{\mathbf{X}|\mathbf{U}}$ (instead of the unknown optimal decoder $P_{\mathbf{X}|\mathbf{U}}$), let $\mathbf{Q} := \{Q_{\mathbf{X}|\mathbf{U}}, Q_{\mathbf{U}}\}$. Also, let

$$\mathcal{L}_s^{\mathrm{VB}}(\mathbf{P}, \mathbf{Q}) := \mathbb{E}_{P_{\mathbf{X}}} \left[ \mathbb{E}_{P_{\mathbf{U}|\mathbf{X}}}[\log Q_{\mathbf{X}|\mathbf{U}}] - sD_{\mathrm{KL}}(P_{\mathbf{U}|\mathbf{X}} \| Q_{\mathbf{U}}) \right] . \tag{6}$$

**Lemma 1.** *For given* $\mathbf{P}$*, we have*

$$\mathcal{L}_s^{\mathrm{VB}}(\mathbf{P}, \mathbf{Q}) \leq \tilde{\mathcal{L}}_s(\mathbf{P}), \quad \text{for all } \mathbf{Q} .$$

*In addition, there exists a unique* $\mathbf{Q}$ *that achieves the maximum* $\max_{\mathbf{Q}} \mathcal{L}_s^{\mathrm{VB}}(\mathbf{P}, \mathbf{Q}) = \tilde{\mathcal{L}}_s(\mathbf{P})$*, and is given by*

$$Q_{\mathbf{X}|\mathbf{U}}^* = P_{\mathbf{X}|\mathbf{U}} , \quad Q_{\mathbf{U}}^* = P_{\mathbf{U}} . \qquad \square$$

Using Lemma 1, maximization of equation 4 can be written in term of the variational IB cost as follows

$$\max_{\mathbf{P}} \mathcal{L}_s'(\mathbf{P}) = \max_{\mathbf{P}} \max_{\mathbf{Q}} \mathcal{L}_s^{\mathrm{VB}}(\mathbf{P}, \mathbf{Q}) . \tag{7}$$

**Remark 1.** *As we already mentioned in the beginning of this chapter, the related work (Jiang et al., 2017) performs unsupervised clustering by combining VAE with GMM. Specifically, it maximizes the following ELBO bound*

$$\mathcal{L}_1^{\mathrm{VaDE}} := \mathbb{E}_{P_{\mathbf{X}}} \left[ \mathbb{E}_{P_{\mathbf{U}|\mathbf{X}}}[\log Q_{\mathbf{X}|\mathbf{U}}] - D_{\mathrm{KL}}(P_{C|\mathbf{X}} \| Q_C) - \mathbb{E}_{P_{C|\mathbf{X}}}[D_{\mathrm{KL}}(P_{\mathbf{U}|\mathbf{X}} \| Q_{\mathbf{U}|C})] \right] . \tag{8}$$

*Let, for an arbitrary non-negative parameter* $s$*,* $\mathcal{L}_s^{\mathrm{VaDE}}$ *be a generalization of the ELBO bound in equation 8 of (Jiang et al., 2017) given by*

$$\mathcal{L}_s^{\mathrm{VaDE}} := \mathbb{E}_{P_{\mathbf{X}}} \left[ \mathbb{E}_{P_{\mathbf{U}|\mathbf{X}}}[\log Q_{\mathbf{X}|\mathbf{U}}] - sD_{\mathrm{KL}}(P_{C|\mathbf{X}} \| Q_C) - s\mathbb{E}_{P_{C|\mathbf{X}}}[D_{\mathrm{KL}}(P_{\mathbf{U}|\mathbf{X}} \| Q_{\mathbf{U}|C})] \right] . \tag{9}$$

*Investigating the RHS of equation 9, we get*

$$\mathcal{L}_s^{\mathrm{VB}}(\mathbf{P}, \mathbf{Q}) = \mathcal{L}_s^{\mathrm{VaDE}} + s\mathbb{E}_{P_{\mathbf{X}}} \left[ \mathbb{E}_{P_{\mathbf{U}|\mathbf{X}}}[D_{\mathrm{KL}}(P_{C|\mathbf{X}} \| Q_{C|\mathbf{U}})] \right] . \tag{10}$$

*Thus, by the non-negativity of relative entropy it is clear that* $\mathcal{L}_s^{\mathrm{VaDE}}$ *is a lower bound on* $\mathcal{L}_s^{\mathrm{VB}}(\mathbf{P}, \mathbf{Q})$*. Also, if variational distribution* $\mathbf{Q}$ *is such that the conditional marginal* $Q_{C|\mathbf{U}}$ *is equal to* $P_{C|\mathbf{X}}$ *the bound is tight since the relative entropy term is zero in this case.*

### 3.2 Proposed algorithm: VIB-GMM

In order to compute equation 7, we parametrize the distributions $P_{\mathbf{U}|\mathbf{X}}$ and $Q_{\mathbf{X}|\mathbf{U}}$ using DNNs. For instance, let the stochastic encoder $P_{\mathbf{U}|\mathbf{X}}$ be a DNN $f_\theta$ and the stochastic decoder $Q_{\mathbf{X}|\mathbf{U}}$ be a DNN $g_\phi$. That is

$$\begin{aligned} P_\theta(\mathbf{u}|\mathbf{x}) &= \mathcal{N}(\mathbf{u}; \boldsymbol{\mu}_\theta, \boldsymbol{\Sigma}_\theta) , \quad \text{where } [\boldsymbol{\mu}_\theta, \boldsymbol{\Sigma}_\theta] = f_\theta(\mathbf{x}) , \\ Q_\phi(\mathbf{x}|\mathbf{u}) &= g_\phi(\mathbf{u}) = [\hat{\mathbf{x}}] , \end{aligned} \tag{11}$$

where $\theta$ and $\phi$ are the weight and bias parameters of the DNNs. Furthermore, the latent space is modeled as a GMM with $|\mathcal{C}|$ components with parameters $\psi := \{\pi_c, \boldsymbol{\mu}_c, \boldsymbol{\Sigma}_c\}_{c=1}^{|\mathcal{C}|}$, i.e.,

$$Q_\psi(\mathbf{u}) = \sum_c \pi_c \, \mathcal{N}(\mathbf{u}; \boldsymbol{\mu}_c, \boldsymbol{\Sigma}_c) \,. \tag{12}$$

Using the parametrizations above, the optimization of equation 7 can be rewritten as

$$\max_{\theta, \phi, \psi} \, \mathcal{L}_s^{\mathrm{NN}}(\theta, \phi, \psi) \tag{13}$$

where the cost function $\mathcal{L}_s^{\mathrm{NN}}(\theta, \phi, \psi)$ given by

$$\mathcal{L}_s^{\mathrm{NN}}(\theta, \phi, \psi) := \mathbb{E}_{P_\mathbf{X}} \Big[ \mathbb{E}_{P_\theta(\mathbf{U}|\mathbf{X})}[\log Q_\phi(\mathbf{X}|\mathbf{U})] - s D_{\mathrm{KL}}(P_\theta(\mathbf{U}|\mathbf{X}) \| Q_\psi(\mathbf{U})) \Big] \,. \tag{14}$$

Then, for a given observations of $N$ samples, i.e., $\{\mathbf{x}_i\}_{i=1}^N$, equation 13 can be approximated in terms of an empirical cost as follows

$$\max_{\theta, \phi, \psi} \, \frac{1}{n} \sum_{i=1}^n \mathcal{L}_{s,i}^{\mathrm{emp}}(\theta, \phi, \psi) \,, \tag{15}$$

where $\mathcal{L}_{s,i}^{\mathrm{emp}}(\theta, \phi, \psi)$ is the empirical cost for the $i$-th observation $\mathbf{x}_i$, and given by

$$\mathcal{L}_{s,i}^{\mathrm{emp}}(\theta, \phi, \psi) = \mathbb{E}_{P_\theta(\mathbf{U}_i|\mathbf{X}_i)}[\log Q_\phi(\mathbf{X}_i|\mathbf{U}_i)] - s D_{\mathrm{KL}}(P_\theta(\mathbf{U}_i|\mathbf{X}_i) \| Q_\psi(\mathbf{U}_i)) \,. \tag{16}$$

Furthermore, the first term of the RHS of equation 16 can be computed using Monte Carlo sampling and the re-parametrization trick (Kingma & Welling, 2014). In particular, $P_\theta(\mathbf{u}|\mathbf{x})$ can be sampled by first sampling a random variable $\mathbf{Z}$ with distribution $P_\mathbf{Z}$, i.e., $P_\mathbf{Z} = \mathcal{N}(\mathbf{0}, \mathbf{I})$, then transforming the samples using some function $\tilde{f}_\theta : \mathcal{X} \times \mathcal{Z} \to \mathcal{U}$, i.e., $\mathbf{u} = \tilde{f}_\theta(\mathbf{x}, \mathbf{z})$. Thus,

$$\mathbb{E}_{P_\theta(\mathbf{U}_i|\mathbf{X}_i)}[\log Q_\phi(\mathbf{X}_i|\mathbf{U}_i)] = \frac{1}{M} \sum_{m=1}^M \log q(\mathbf{x}_i|\mathbf{u}_{i,m}), \quad \mathbf{u}_{i,m} = \boldsymbol{\mu}_{\theta,i} + \boldsymbol{\Sigma}_{\theta,i}^{\frac{1}{2}} \cdot \boldsymbol{\epsilon}_m, \quad \boldsymbol{\epsilon}_m \sim \mathcal{N}(\mathbf{0}, \mathbf{I}) \,,$$

where $M$ is the number of samples for the Monte Carlo sampling step.

The second term of the RHS of equation 16 is the KL divergence between a single component multivariate Gaussian and a Gaussian Mixture Model with $|\mathcal{C}|$ components. An exact closed-form solution for the calculation of this term does not exist. However, a variational lower bound approximation (Hershey & Olsen, 2007) of it can be obtained as

$$D_{\mathrm{KL}}(P_{\boldsymbol{\theta}}(\mathbf{U}_i|\mathbf{X}_i) \| Q_{\boldsymbol{\psi}}(\mathbf{U}_i)) = -\log \sum_{c=1}^{|\mathcal{C}|} \pi_c \exp\left(-D_{\mathrm{KL}}(\mathcal{N}(\boldsymbol{\mu}_{\theta,i}, \boldsymbol{\Sigma}_{\theta,i}) \| \mathcal{N}(\boldsymbol{\mu}_c, \boldsymbol{\Sigma}_c))\right) \,. \tag{17}$$

In particular, in the specific case in which the covariance matrices are diagonal, i.e., $\boldsymbol{\Sigma}_{\theta,i} := \mathrm{diag}(\{\sigma_{\theta,i,j}^2\}_{j=1}^{n_u})$ and $\boldsymbol{\Sigma}_c := \mathrm{diag}(\{\sigma_{c,j}^2\}_{j=1}^{n_u})$, with $n_u$ denoting the latent space dimension, equation 17 can be computed as follows

$$D_{\mathrm{KL}}(P_{\boldsymbol{\theta}}(\mathbf{U}_i|\mathbf{X}_i) \| Q_{\boldsymbol{\psi}}(\mathbf{U}_i))$$
$$= -\log \sum_{c=1}^{|\mathcal{C}|} \pi_c \exp\left(-\frac{1}{2} \sum_{j=1}^{n_u} \left[\frac{(\mu_{\theta,i,j} - \mu_{c,j})^2}{\sigma_{c,j}^2} + \log \frac{\sigma_{c,j}^2}{\sigma_{\theta,i,j}^2} - 1 + \frac{\sigma_{\theta,i,j}^2}{\sigma_{c,j}^2}\right]\right) \,, \tag{18}$$

where $\mu_{\theta,i,j}$ and $\sigma_{\theta,i,j}^2$ are the mean and variance of the $i$-th representation in the $j$-th dimension of the latent space. Furthermore, $\mu_{c,j}$ and $\sigma_{c,j}^2$ represent the mean and variance of the $c$-th component of the GMM in the $j$-th dimension of the latent space.

Finally, we train NNs to maximize the cost function equation 14 over the parameters $\theta, \phi$, as well as those $\psi$ of the GMM. For the training step, we use the ADAM optimization tool (Kingma & Ba, 2015). The training procedure is detailed in Algorithm 1.

Once our model is trained, we assign the given dataset into the clusters. As mentioned in Section 2, we do the assignment from the latent representations, i.e., $Q_{C|\mathbf{U}} = P_{C|\mathbf{X}}$. Hence, the probability that the observed data $\mathbf{x}_i$ belongs to the $c$-th cluster is computed as follows

$$p(c|\mathbf{x}_i) = q(c|\mathbf{u}_i) = \frac{q_{\psi^\star}(c) q_{\psi^\star}(\mathbf{u}_i|c)}{q_{\psi^\star}(\mathbf{u}_i)} = \frac{\pi_c^\star \mathcal{N}(\mathbf{u}_i; \boldsymbol{\mu}_c^\star, \boldsymbol{\Sigma}_c^\star)}{\sum_c \pi_c^\star \mathcal{N}(\mathbf{u}_i; \boldsymbol{\mu}_c^\star, \boldsymbol{\Sigma}_c^*)} \ , \tag{19}$$

where $^\star$ indicates optimal values of the parameters as found at the end of the training phase. Finally, the right cluster is picked based on the largest assignment probability value.

---

**Algorithm 1** VIB-GMM algorithm for unsupervised learning

---

1: **input:** Dataset $\mathcal{D} := \{\mathbf{x}_i\}_{i=1}^n$, parameter $s \geq 0$.
2: **output:** Optimal DNN weights $\theta^\star, \phi^\star$ and GMM parameters $\psi^\star = \{\pi_c^\star, \boldsymbol{\mu}_c^\star, \boldsymbol{\Sigma}_c^\star\}_{c=1}^{|\mathcal{C}|}$.
3: **initialization** Initialize $\theta, \phi, \psi$.
4: **repeat**
5:     Randomly select $b$ mini-batch samples $\{\mathbf{x}_i\}_{i=1}^b$ from $\mathcal{D}$.
6:     Draw $m$ random i.i.d samples $\{\mathbf{z}_j\}_{j=1}^m$ from $P_\mathbf{Z}$.
7:     Compute $m$ samples $\mathbf{u}_{i,j} = \tilde{f}_\theta(\mathbf{x}_i, \mathbf{z}_j)$
8:     For the selected mini-batch, compute gradients of the empirical cost equation 15.
9:     Update $\theta, \phi, \psi$ using the estimated gradient (e.g. with SGD or ADAM).
10: **until** convergence of $\theta, \phi, \psi$.

---

**Remark 2.** *It is worth to mention that with the use of the KL approximation in equation 17, our algorithm does not use the assumption $P_{C|\mathbf{U}} = Q_{C|\mathbf{U}}$ (not as in Jiang et al. (2017)), which does not hold in the beginning of the training phase and leads to convergence issues. This assumption is only used in the final assignment after the training phase is over.*

### 3.3 EFFECT OF THE HYPER-PARAMETER

As we already mentioned, the hyper-parameter $s$ controls the trade-off between the relevance of the representation $\mathbf{U}$ and its complexity. As it can be seen from equation 14 for small values of $s$, it is the cross-entropy term that dominates, i.e., the algorithm trains the parameters so as to reproduce $\mathbf{X}$ as accurate as possible. For large values of $s$, however, it is most important for the NN to produce an encoded version of $\mathbf{X}$ whose distribution matches the prior distribution of the latent space, i.e., the term $D_{\mathrm{KL}}(P_{\boldsymbol{\theta}}(\mathbf{U}|\mathbf{X}) \| Q_\psi(\mathbf{U}))$ is nearly zero.

---

**Algorithm 2** Annealing Algorithm Pseudo-Code

---

**input:** Dataset $\mathcal{D} := \{\mathbf{x}_i\}_{i=1}^n$,
    hyper-parameter interval $[s_{\min}, s_{\max}]$.
**output:** Optimal DNN weights $\theta^\star, \phi^\star$, GMM
    parameters $\psi^\star = \{\pi_c^\star, \boldsymbol{\mu}_c^\star, \boldsymbol{\Sigma}_c^\star\}_{c=1}^{|\mathcal{C}|}$,
    assignment probability $P_{C|\mathbf{X}}$.
**initialization** Initialize $\theta, \phi, \psi$.
**repeat**
    Apply VIB-GMM algorithm.
    Update $\psi, \theta, \phi$.
    Update $s$, e.g., $s = (1 + \epsilon_s)s_{\mathrm{old}}$.
**until** $s$ does not exceed $s_{\max}$.

---

In the beginning of the training process, the GMM components are randomly selected; and so starting with a large value of the hyper-parameter $s$ is likely to steer the solution towards an irrelevant prior. Hence, for the tunning of the hyper-parameter $s$ in practice it is more efficient to start with a small value of $s$ and gradually increase it with the number of epochs. This has the advantage to avoid possible local minimas, an aspect that is reminiscent of deterministic annealing (Slonim, 2002), where $s$ plays the role of the temperature parameter. The experiments that will be reported in the next section show that proceeding in the above described manner for the selection of the parameter $s$ helps getting better accuracy results and better robustness to the initialization (i.e., no need for a strong pretraining). A pseudo-code for annealing is given in Algorithm 2. We note that tuning $s$ is very critical, such that the step size $\epsilon_s$ in update of $s$ should be chosen carefully, otherwise phase transitions might be skipped that would cause a bad ACC score.

## 4 EXPERIMENTS

### 4.1 DESCRIPTION OF USED DATASETS

In our empirical experiments, we apply our algorithm to the clustering of the following datasets.

**MNIST**: A dataset of gray-scale images of 70000 handwritten digits of dimensions $28 \times 28$ pixel.

**STL-10**: A dataset of color images collected from 10 categories. Each category consists of 1300 images of size of $96 \times 96$ (pixels) $\times 3$ (rgb code). Hence, the original input dimension $n_x$ is 27648. For this dataset, we use a pretrained convolutional NN model, i.e., ResNet-50 (He et al., 2016) to reduce the dimensionality of the input. This preprocessing reduces the input dimension to 2048. Then, our algorithm and other baselines are used for clustering.

**REUTERS10K**: A dataset that is composed of 810000 English stories labeled with a category tree. As in (Xie et al., 2016), 4 root categories (corporate/industrial, government/social, markets, economics) are selected as labels and all documents with multiple labels are discarded. Then, tf-idf features are computed on the 2000 most frequently occurring words. Finally, 10000 samples are taken randomly, which are referred to as REUTERS10K dataset.

## 4.2 NETWORK SETTINGS AND OTHER PARAMETERS

We use the following network architecture: the encoder is modeled with NNs with 3 hidden layers with dimensions $n_x - 500 - 500 - 2000 - J$, where $n_x$ is the input dimension and $n_u$ is the dimension of the latent space. The decoder consists of NNs with dimensions $n_u - 2000 - 500 - 500 - n_x$. All layers are fully connected. For comparison purposes, we chose the architecture of the hidden layers as well as the dimension of the latent space $n_u = 10$ to coincide with those made for the DEC algorithm of (Xie et al., 2016) and the VaDE algorithm of (Jiang et al., 2017). All except the last layers of the encoder and decoder are activated with ReLU function. For the last (i.e., latent) layer of the encoder we use a linear activation; and for the last (i.e., output) layer of the decoder we use sigmoid function for MNIST and linear activation for the remaining datasets. The batch size is 100 and the variational bound equation 15 is maximized by the Adam optimizer of (Kingma & Ba, 2015). The learning rate is initialized with 0.002 and decreased gradually every 20 epochs with a decay rate of 0.9 until it reaches a small value (0.0005 is our experiments). The reconstruction loss is calculated by using the cross-entropy criterion for MNIST and mean squared error function for the other datasets.

|         | MNIST | STL-10 | REUTERS10K |
|---------|-------|--------|------------|
| GMM     | 50.4  | 77.1   | 53.74      |
| DEC     | 84.3[‡] | 80.6[†] | 72.17[‡]   |
| VaDE    | 94.5[†] | 84.3   | 79.8[†]    |
| **VIB-GMM** | **96.2** | **91.6** | **80.4** |

[†] values are taken from VaDE (Jiang et al., 2017)
[‡] values are taken from DEC (Xie et al., 2016)

Table 1: Comparison of clustering accuracy of various algorithms.

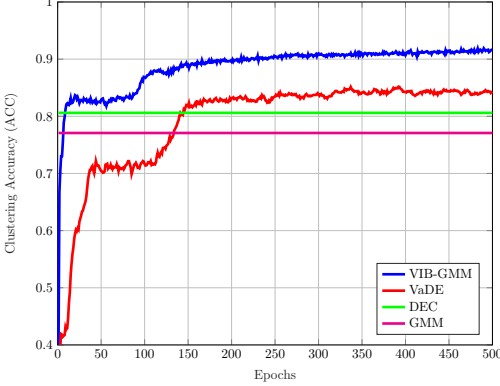
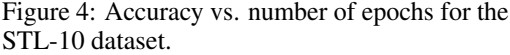

Figure 4: Accuracy vs. number of epochs for the STL-10 dataset.

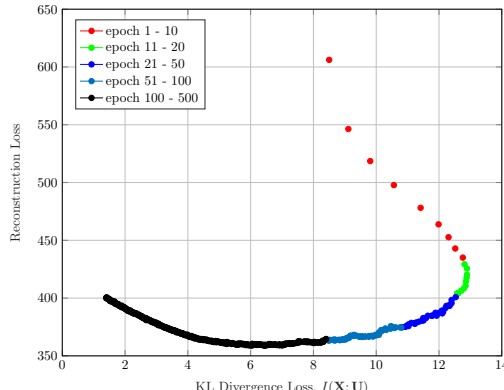

Figure 5: Information plane for the STL-10 dataset.

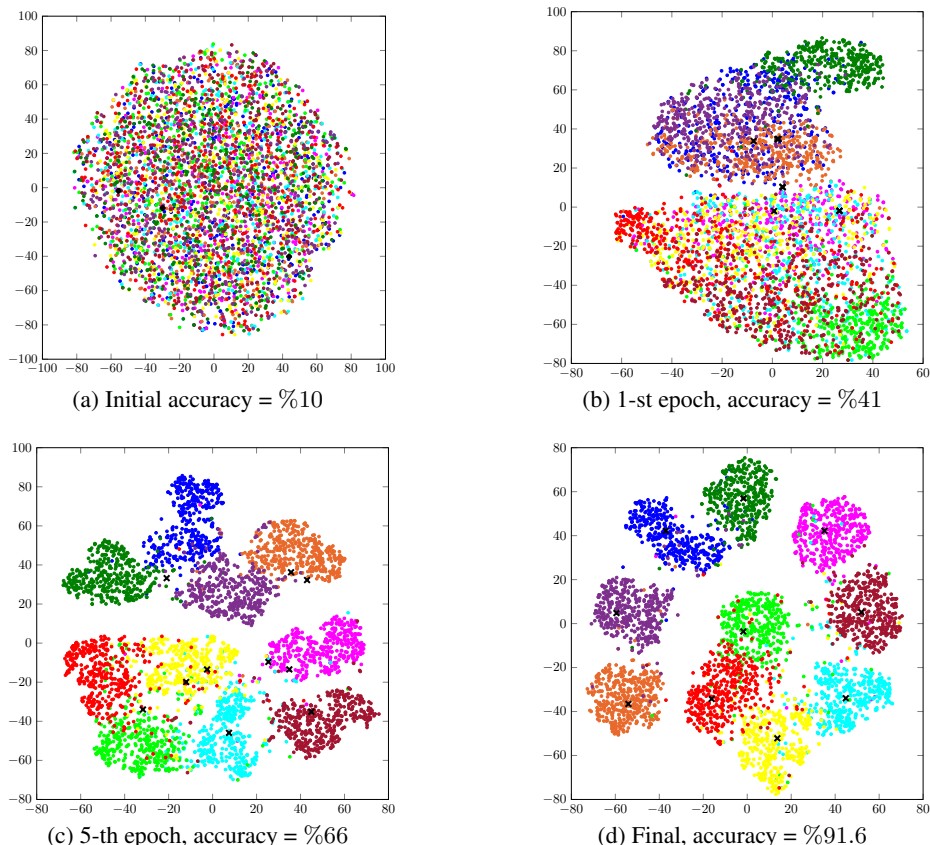

(a) Initial accuracy = %10

(b) 1-st epoch, accuracy = %41

(c) 5-th epoch, accuracy = %66

(d) Final, accuracy = %91.6

Figure 6: Visualization of the latent space before training; and after 1, 5 and 500 epochs.

## 4.3 CLUSTERING ACCURACY

We evaluate the performance of our algorithm in terms of the so-called unsupervised clustering accuracy (ACC), which is a widely used metric in the context of unsupervised learning (Min et al., 2018). For comparison purposes, we also present those of algorithms from previous art.

For each of the aforementioned datasets, we run our VIB-GMM algorithm for various values of the hyper-parameter $s$ inside an interval $[s_{\min}, s_{\max}]$, starting from the smaller valuer $s_1$ and gradually increasing the value of $s$ every $n_{\text{epoch}}$ epochs. For the MNIST dataset, we set $(s_{\min}, s_{\max}, n_{\text{epoch}}) = (1, 5, 500)$; and for the STL-10 dataset and the REUTERS10K datset we choose these parameters to be $(1, 20, 500)$ and $(1, 5, 100)$, respectively. The obtained ACC accuracy results are reported in the Table 1 from which it can be seen that our algorithm outperforms significantly the DEC algorithm of (Xie et al., 2016) as well as the VaDE algorithm of (Jiang et al., 2017) and GMM on the same datsets. Important to note, for the MNIST dataset the reported ACC accuracy of $96.2\%$ using our VIB-GMM algorithm is obtained as the best case run out of ten times run all with random initializations. For instance, we do not use any pretrained values for the initialization of our algorithm in sharp contrast with the VaDE of (Jiang et al., 2017) and the DEC of (Xie et al., 2016). For the STL-10 dataset, none of the compared algorithms use a pretrained network except the intimal ResNet-50 for dimensionality reduction. For REUTERS10K, we used the same pretrain parameters as DEC and VaDE. Figure 4 depicts the evolution of the ACC accuracy with iterations (number of epochs) for the four compared algorithms.

Figure 5 shows the evolution of the reconstruction loss of our VIB-GMM algorithm for the STL-10 dataset, as a function of simultaneously varying values of the hyper-parameter $s$ and the number of epochs (recall that, as per-the described methodology, we start with $s = s_1$ and we increase its value gradually every $n_{\text{epoch}} = 500$ epochs). As it can be seen from the figure, the few first epochs are spent

almost entirely on reducing the reconstruction loss (i.e., a fitting phase) and most of the remaining epochs are spent in making the found representation more concise (i.e., smaller KL-divergence). This is reminiscent of the two-phase (fitting v.s. compression) that was observed for supervised learning using VIB in (Schwartz-Ziv & Tishby, 2017).

### 4.4 VISUALIZATION ON THE LATENT SPACE

In this section, we investigate the evolution of the unsupervised clustering of the STL-10 dataset on the latent space using our VIB-GMM algorithm. For this purpose, we find it convenient to visualize the latent space through application of the t-SNE algorithm of (van der Maaten & Hinton, 2008) in order to generate meaningful representations in a two-dimensional space. Figure 6 shows $4000$ randomly chosen latent representations before the start of the training process and respectively after $1$, $5$ and $500$ epochs. The shown points (with a $\cdot$ marker in the figure) represent latent representations of data samples whose labels are identical. Colors are used to distinguish between clusters. Crosses (with an $\mathbf{x}$ marker in the figure) correspond to the centroids of the clusters. More specifically, Figure 6-(a) shows the initial latent space before the training process. If the clustering is performed on the initial representations it allows ACC accuracy of as small as $10\%$, i.e., as bad as a random assignment. Figure 6-(b) shows the latent space after one epoch, from which a partition of some of the points starts to be already visible. With five epochs, that partitioning is significantly sharper and the associated clusters can be recognized easily. Observe, however, that the cluster centers seem still not to have converged. With $500$ epochs, the ACC accuracy of our algorithm reaches $\%91.6$ and the clusters and their centroids are neater as visible from Figure 6-(d).

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

## A   THE PROOF OF LEMMA 1

First, we expand $\tilde{\mathcal{L}}_s(\mathbf{P})$ as follows

$$
\begin{aligned}
\tilde{\mathcal{L}}_s(\mathbf{P}) = & -H(\mathbf{X}|\mathbf{U}) - sI(\mathbf{X};\mathbf{U}) \\
= & -H(\mathbf{X}|\mathbf{U}) - s[H(\mathbf{U}) - H(\mathbf{U}|\mathbf{X})] \\
= & \iint_{\mathbf{ux}} p(\mathbf{u},\mathbf{x}) \log p(\mathbf{x}|\mathbf{u}) \, d\mathbf{u} \, d\mathbf{x} \\
& + s \int_{\mathbf{u}} p(\mathbf{u}) \log p(\mathbf{u}) \, d\mathbf{u} - s \iint_{\mathbf{ux}} p(\mathbf{u},\mathbf{x}) \log p(\mathbf{u}|\mathbf{x}) \, d\mathbf{u} \, d\mathbf{x}.
\end{aligned}
$$

Then, $\mathcal{L}_s^{\mathrm{VB}}(\mathbf{P},\mathbf{Q})$ is defined as follows

$$
\begin{aligned}
\mathcal{L}_s^{\mathrm{VB}}(\mathbf{P},\mathbf{Q}) := & \iint_{\mathbf{ux}} p(\mathbf{u},\mathbf{x}) \log q(\mathbf{x}|\mathbf{u}) \, d\mathbf{u} \, d\mathbf{x} \\
& + s \int_{\mathbf{u}} p(\mathbf{u}) \log q(\mathbf{u}) \, d\mathbf{u} - s \iint_{\mathbf{ux}} p(\mathbf{u},\mathbf{x}) \log p(\mathbf{u}|\mathbf{x}) \, d\mathbf{u} \, d\mathbf{x}.
\end{aligned}
\tag{20}
$$

Hence, we have the following relation

$$\tilde{\mathcal{L}}_s(\mathbf{P}) - \mathcal{L}_s^{\mathrm{VB}}(\mathbf{P}, \mathbf{Q}) = \mathbb{E}_{P_{\mathbf{X}}}[D_{\mathrm{KL}}(P_{\mathbf{X}|\mathbf{U}}\|Q_{\mathbf{X}|\mathbf{U}})] + sD_{\mathrm{KL}}(P_{\mathbf{U}}\|Q_{\mathbf{U}}) \geq 0$$

where equality holds under equalities $Q_{\mathbf{X}|\mathbf{U}} = P_{\mathbf{X}|\mathbf{U}}$ and $Q_{\mathbf{U}} = P_{\mathbf{U}}$. We note that $s \geq 0$.

# B  THE PROOF OF ALTERNATIVE EXPRESSION $\mathcal{L}_s^{\mathrm{VADE}}$

Here we show how we obtained equation 10.

To do so,

$$\mathcal{L}_s^{\mathrm{VaDE}} = \mathbb{E}_{P_{\mathbf{X}}}\Big[\mathbb{E}_{P_{\mathbf{U}|\mathbf{X}}}[\log Q_{\mathbf{X}|\mathbf{U}}] - sD_{\mathrm{KL}}(P_{\mathbf{U}|\mathbf{X}}\|Q_{\mathbf{U}}) - s\mathbb{E}_{P_{\mathbf{U}|\mathbf{X}}}\big[D_{\mathrm{KL}}(P_{C|\mathbf{X}}\|Q_{C|\mathbf{U}})\big]\Big]$$

$$= \mathbb{E}_{P_X}\big[\mathbb{E}_{P_{\mathbf{U}|\mathbf{X}}}[\log Q_{\mathbf{X}|\mathbf{U}}]\big] - s\int_{\mathbf{x}} p(\mathbf{x}) \int_{\mathbf{u}} p(\mathbf{u}|\mathbf{x}) \log \frac{p(\mathbf{u}|\mathbf{x})}{q(\mathbf{u})}\, d\mathbf{u}\, d\mathbf{x}$$

$$- s\int_{\mathbf{x}} p(\mathbf{x}) \int_{\mathbf{u}} p(\mathbf{u}|\mathbf{x}) \sum_c p(c|\mathbf{x}) \log \frac{p(c|\mathbf{x})}{q(c|\mathbf{u})}\, d\mathbf{u}\, d\mathbf{x}$$

$$\overset{(a)}{=} \mathbb{E}_{P_X}\big[\mathbb{E}_{P_{\mathbf{U}|\mathbf{X}}}[\log Q_{\mathbf{X}|\mathbf{U}}]\big] - s\iint_{\mathbf{ux}} p(\mathbf{x})p(\mathbf{u}|\mathbf{x}) \log \frac{p(\mathbf{u}|\mathbf{x})}{q(\mathbf{u})}\, d\mathbf{u}\, d\mathbf{x}$$

$$- s\iint_{\mathbf{ux}} \sum_c p(\mathbf{x})p(\mathbf{u}|c,\mathbf{x})p(c|\mathbf{x}) \log \frac{p(c|\mathbf{x})}{q(c|\mathbf{u})}\, d\mathbf{u}\, d\mathbf{x}$$

$$= \mathbb{E}_{P_X}\big[\mathbb{E}_{P_{\mathbf{U}|\mathbf{X}}}[\log Q_{\mathbf{X}|\mathbf{U}}]\big] - s\iint_{\mathbf{ux}} \sum_c p(\mathbf{u},c,\mathbf{x}) \log \frac{p(\mathbf{u}|\mathbf{x})p(c|\mathbf{x})}{q(\mathbf{u})q(c|\mathbf{u})}\, d\mathbf{u}\, d\mathbf{x}$$

$$= \mathbb{E}_{P_X}\big[\mathbb{E}_{P_{\mathbf{U}|\mathbf{X}}}[\log Q_{\mathbf{X}|\mathbf{U}}]\big] - s\iint_{\mathbf{ux}} \sum_c p(\mathbf{u},c,\mathbf{x}) \log \frac{p(c|\mathbf{x})}{q(c)} \frac{p(\mathbf{u}|\mathbf{x})}{q(\mathbf{u}|c)}\, d\mathbf{u}\, d\mathbf{x}$$

$$= \mathbb{E}_{P_X}\big[\mathbb{E}_{P_{\mathbf{U}|\mathbf{X}}}[\log Q_{\mathbf{X}|\mathbf{U}}]\big] - s\int_{\mathbf{x}} \sum_c p(c,\mathbf{x}) \log \frac{p(c|\mathbf{x})}{q(c)}\, d\mathbf{x}$$

$$- s\iint_{\mathbf{ux}} \sum_c p(\mathbf{x})p(c|\mathbf{x})p(\mathbf{u}|c,\mathbf{x}) \log \frac{p(\mathbf{u}|\mathbf{x})}{q(\mathbf{u}|c)}\, d\mathbf{u}\, d\mathbf{x}$$

$$\overset{(b)}{=} \mathbb{E}_{P_{\mathbf{X}}}\Big[\mathbb{E}_{P_{\mathbf{U}|\mathbf{X}}}[\log Q_{\mathbf{X}|\mathbf{U}}] - sD_{\mathrm{KL}}(P_{C|\mathbf{X}}\|Q_C) - s\mathbb{E}_{P_{C|\mathbf{X}}}[D_{\mathrm{KL}}(P_{\mathbf{U}|\mathbf{X}}\|Q_{\mathbf{U}|C})]\Big]$$

$$\overset{(c)}{=} \mathcal{L}_s^{\mathrm{VB}}(\mathbf{P}, \mathbf{Q}) - s\mathbb{E}_{P_{\mathbf{X}}}\Big[\mathbb{E}_{P_{\mathbf{U}|\mathbf{X}}}\big[D_{\mathrm{KL}}(P_{C|\mathbf{X}}\|Q_{C|\mathbf{U}})\big]\Big],$$

where $(a)$ and $(b)$ follow due to the Markov chain $C \!\!-\!\!\circ\!\!-\!\! \mathbf{X} \!\!-\!\!\circ\!\!-\!\! \mathbf{U}$; $(c)$ follows from the definition of $\mathcal{L}_s^{\mathrm{VB}}(\mathbf{P}, \mathbf{Q})$ in equation 6.

# C  KL DIVERGENCE BETWEEN MULTIVARIATE GAUSSIAN DISTRIBUTIONS

The KL divergence between two multivariate Gaussian distributions $P_1 \sim \mathcal{N}(\boldsymbol{\mu}_1, \boldsymbol{\Sigma}_1)$ and $P_2 \sim \mathcal{N}(\boldsymbol{\mu}_2, \boldsymbol{\Sigma}_2)$ in $\mathbb{R}^J$ is

$$D_{\mathrm{KL}}(P_1\|P_2) = \frac{1}{2}\Big((\boldsymbol{\mu}_1 - \boldsymbol{\mu}_2)^T \boldsymbol{\Sigma}_2^{-1}(\boldsymbol{\mu}_1 - \boldsymbol{\mu}_2) + \log|\boldsymbol{\Sigma}_2| - \log|\boldsymbol{\Sigma}_1| - J + \mathrm{tr}(\boldsymbol{\Sigma}_2^{-1}\boldsymbol{\Sigma}_1)\Big). \quad (21)$$

For the case in which $\boldsymbol{\Sigma}_1$ and $\boldsymbol{\Sigma}_2$ covariance matrices are diagonal, i.e., $\boldsymbol{\Sigma}_1 := \mathrm{diag}(\{\sigma_{1,j}^2\}_{j=1}^J)$ and $\boldsymbol{\Sigma}_2 := \mathrm{diag}(\{\sigma_{2,j}^2\}_{j=1}^J)$, equation 21 boils down to the following

$$D_{\mathrm{KL}}(P_1\|P_2) = \frac{1}{2}\Big(\sum_{j=1}^J \frac{(\mu_{1,j} - \mu_{2,j})^2}{\sigma_{2,j}^2} + \log\frac{\sigma_{2,j}^2}{\sigma_{1,j}^2} - 1 + \frac{\sigma_{1,j}^2}{\sigma_{2,j}^2}\Big). \quad (22)$$

# D    KL Divergence Between Gaussian Mixture Models

An exact close form for the calculation of the KL divergence between two Gaussian mixture models does not exist. In this paper, we use a variational lower bound approximation for calculations of KL between two Gaussian mixture models. Let $f$ and $g$ be GMMs and the marginal densities of $x$ under $f$ and $g$ are

$$f(x) = \sum_{m=1}^{M} \omega_m \mathcal{N}(x; \mu_m^{\mathrm{f}}, \Sigma_m^{\mathrm{f}}) = \sum_{m=1}^{M} \omega_m f_m(x)$$

$$g(x) = \sum_{C=1}^{C} \pi_c \mathcal{N}(x; \mu_c^{\mathrm{g}}, \Sigma_c^{\mathrm{g}}) = \sum_{c=1}^{C} \pi_c g_c(x).$$

The KL divergence between two Gaussian mixtures f an g can be approximated as follows

$$D_{\mathrm{vKL}}(f\|g) := \sum_{m=1}^{M} \omega_m \log \frac{\sum_{m' \in \mathcal{M}\backslash m} \omega_{m'} \exp\left(-D_{\mathrm{KL}}(f_m\|f_{m'})\right)}{\sum_{c=1}^{C} \pi_c \exp\left(-D_{\mathrm{KL}}(f_m\|g_c)\right)}. \tag{23}$$

In this paper, we are interested, in particular, $M = 1$. Hence, equation 23 simplifies to

$$D_{\mathrm{vKL}}(f\|g) = -\log \sum_{c=1}^{C} \pi_c \exp\left(-D_{\mathrm{KL}}(f\|g_c)\right), \tag{24}$$

where $D_{\mathrm{KL}}(\cdot\|\cdot)$ is the KL divergence between single component multivariate Gaussian distribution, defined as in equation 21.

