# OpenReview forum: "Variational Information Bottleneck for Unsupervised Clustering: Deep Gaussian Mixture Embedding"
_ICLR.cc/2020/Conference — Reject_

### Official Review · AnonReviewer2 · 2019-10-23
**Official Blind Review #2**

**Rating:** 3

**Review:**

This paper considers the autoencoder model combining the usual information bottleneck and the Gaussian mixture model (GMM). Using an approximation to deal with GMMs, the authors derive a bound on the cost function generalizing the ELBO. The performance of the proposed method is tested on three benchmark datasets and compared with existing methods combining VAE with GMM.

While the framework and the performance of the proposed method are interesting and promising, some of its main parts are unclearly explained.

- Although Remark 1 well explains the difference between VaDE and the proposed objective functions, little is discussed how this difference affects the learnt model.
- I wonder why Q_\phi(x|u) in Eq. (11) doesn’t have to be a distribution. What does the expression [\hat{x}] mean?
- Eq. (17) is not explained clearly. Since the equality symbol is used, it is unclear where is approximation. Although this approximation is one of the main parts of the proposed method, little is discussed on the influence of this approximation.
- Are the information plane and latent representations in Figs 5 and 6 also available for DEC and VaDE and not limited to the proposed method?

Minor comments:
p.5, the math expression between Eqs. (16) and (17): The distribution q(x_i|u_i,m) is undefined.
p.6, l.4 from the bottom: ACC is defined later.
p.7, l.14: Should J be n_u?


**Experience Assessment:**

I have read many papers in this area.

**Review Assessment: Checking Correctness Of Derivations And Theory:**

I assessed the sensibility of the derivations and theory.

**Review Assessment: Checking Correctness Of Experiments:**

I assessed the sensibility of the experiments.

**Review Assessment: Thoroughness In Paper Reading:**

I made a quick assessment of this paper.

---

### Official Review · AnonReviewer3 · 2019-10-23
**Official Blind Review #3**

**Rating:** 3

**Review:**

The author(s) posit a Mixture of Gaussian's prior for a compressed latent space representation of high-dimensional data (e.g. images and documents). They propose fitting this model using the Variational Information Bottleneck paradigm and explicate its derivation and tie it to the variational objective used by similar models. They empirically showcase their model and optimization methodology on the MNIST, STL-10, and Reuters10k benchmarks.

The idea of using a latent mixture of Gaussian's to variationally encode high-dimensional data is not new. The author(s) appropriately cite VaDE (Jiang, 2017) and DEC (Xie, 2016), "Deep Unsupervised Clustering with Gaussian Mixture Variational Autoencoders" (Dilokthanakul, 2017). However, the author(s) unfortunately did not mention "Approximate Inference for Deep Latent Gaussian Mixtures" (Nalisnick, 2016). Nalisnick et al. consider the exact generative process as the proposed by the author(s), but with the addition of a Dirichlet prior on the Categorical distribution. Nalisnick et al. fit their model with a VAE and circumvent Dirichlet intractability by positing a Kumarswamy stick-breaking variational posterior (the resulting distribution is on the simplex). They achieve 91.58% accuracy, but do so after fitting a KNN to the latent space. New techniques such as "Implicit Reparameterization Gradients" (Figurnov, 2018) and "Pathwise Derivatives Beyond the Reparameterization Trick" (Jankowiak, 2018) allow direct use of a Dirichlet posterior in VAEs. These methods are respectively implemented in the TensorFlow and PyTorch APIs. My point here is that there are many ways to fit this pre-existing model. From my review of these other works we have, for "best run" on MNIST, that author(s) > VaDE > Nalisnick > Dilokthanakul > DEC. Thus, the author(s) are SoTA for this particular generative process for their best run. It would be nice to see their standard deviation to gauge how statistically significant their results are. However, I am dubious of the SoTA claim as I detail later.

The author(s) derivation was sound, but a bit confusing for me. In particular, I found keeping track of P's and Q's very burdensome after reading sections 2.1 and 2.2. In my experience, Q is typically reserved for a variational distribution that approximates some intractable P distribution, while P is used to describe the generative process (i.e. likelihood) and other exact distributions. Furthermore, one typically introduces the generative process first using P distributions. Once, I got past this confusion everything else made sense. I might suggest introducing the generative process first and with P distributions instead of Q's. The variational model can follow with the Q distributions as the author(s) have it. Equations 4, 5, and 6 all exactly match the unsupervised information bottleneck objective (Alemi, 2017)--see appendix B. I am therefore confident in those equations. I carefully checked their derivation of the VADe comparison (equation 10 and appendix B). Their derivation is straightforward. The principle trick is using the MC assumption C->X->U for P distributions to claim p(u|x,c) = p(u|x). Equations 12-16 all follow naturally from equation 11. If equation 17 is indeed an approximation or bound approximation, I would suggest not using the equals sign. Instead, consider another appropriate operator or rename Dkl to indicate it is the approximated version (just as in equation 24).

If the author(s) like my suggestion regarding generative vs variational nomenclature, I would also change the second sentence of page 6 to something like "We use our variational Qc|u distribution to compute assignments." Thereafter, I would drop the star indicator for optimal parameters. These parameters are not necessarily optimal given the non-convexity of the DNN. Replace with, "after convergence, we ..." Similarly, drop optimal from line 2 of algorithm 1.

Circling back to the experiments, the author(s) use reported values from DEC and VaDE. Those works compute cluster assignments using a KNN classifier on the latent space. This paper however uses the arg max of equation 19. I much prefer this article's method, but for comparison purposes, the author(s) should similarly use a KNN classifier on their latent space to compute accuracy in the same manner. The use of KNN in these other works allows them to consider a number of latent clusters larger than the number of true classes (e.g. 20 clusters for MNIST). I like that the author(s) stick to 10 clusters for MNIST, but for comparison I would have liked to see a KNN generated accuracy alongside their equation 19 based accuracy. It looks like the author(s) implemented VADe for STL-10 based on table 1. If so, it seems they could easily implement equation 19 for their STL-10 value for VADe. If not, please correct this. Not to belabor further, but I would really like to see a table 2 that reports both equation 19 and KNN accuracies when available. Namely, the author(s) should report both values for their model and can leave equation 19 accuracies blank for reported values.

Lastly, I always raise an eyebrow when I see tSNE latent space representations. STL-10 was used to generate figure 6, where one needs dim(u) >> 2. However, MNIST can be well reconstructed using just 2 latent dimensions. In this case, tSNE is unnecessary. The author(s) state "Colors are used to distinguish between clusters." This statement is unclear as to whether the author(s) are using the class label or learned latent cluster. If it is the former, figure 6 makes sense in that it shows misclassifications (i.e. red dots in the green cluster). However, if it is the latter then I am concerned tSNE is doing something weird.

To summarize, I enjoyed the paper. My only concern is novelty. Being the first to pair an existing model with an existing method, in my eyes, does not necessarily meet the ICLR bar. The author(s) seemingly achieve SoTA, but without KNN-based accuracies for their model it is hard to compare to cited works. Having these KNN results and error bars would strengthen their case substantially to: achieving SoTA for an existing model by being the first to pair it with an existing method.


**Experience Assessment:**

I have published one or two papers in this area.

**Review Assessment: Checking Correctness Of Derivations And Theory:**

I carefully checked the derivations and theory.

**Review Assessment: Checking Correctness Of Experiments:**

I carefully checked the experiments.

**Review Assessment: Thoroughness In Paper Reading:**

I read the paper thoroughly.

---

### Official Review · AnonReviewer1 · 2019-10-24
**Official Blind Review #1**

**Rating:** 3

**Review:**

This paper purposes to cluster data in an unsupervised manner that estimates the distribution with GMM in latent space instead of original data space. Also, to better describe the distribution of latent code, the authors involve VIB to constraint the latent code.

The whole pipeline is clear and makes sense. And the experiment proves it's effective.

However, in my eyes, it's almost like an extension of VaDE which combines VIB and GMM too. The main difference is the authors use some hyperparameters to control the optimization of the whole model and make some more proper hypothesis. All of those make sense but may not contribute much to the research field.

**Experience Assessment:**

I have read many papers in this area.

**Review Assessment: Checking Correctness Of Derivations And Theory:**

I did not assess the derivations or theory.

**Review Assessment: Checking Correctness Of Experiments:**

I assessed the sensibility of the experiments.

**Review Assessment: Thoroughness In Paper Reading:**

I read the paper at least twice and used my best judgement in assessing the paper.

---

### Public Comment · ~Pranav_Poduval1 · 2019-09-26
**Great Work, just a simple doubt**

There must be other methods to approximate KL divergence b/w mixture distributions, is Variational the best? The Variational Approximation to KL divergence proof seemed to be true for general distributions, so can ur work be extended to a mixture distributions other than Gaussians, like Mixture of Bernoulli's (just an example)

---

> ### Author Response · Authors · 2019-10-04
> **Reply regarding KL approximation and  extendibility of the model**
>
> Since there is no exact calculation of the KL divergence between two Gaussian mixtures (in our case between a single component multivariate Gaussian and a Gaussian Mixture Model) we have to use an approximation. Our requirements for this approximation are 1) to be closed-form (and differentiable), in order to be able to back-propagate on the cost function by using stochastic gradient descent and 2) the approximation should not depend on the assignment probability $P_{C|X}$ because this probability is not known and as we mentioned in the paper the assumption $P_{C|X} = Q_{C|U}$, affects negatively the training process. The lower bound approximation of Hershey & Olsen (2007) full fills our requirements and works well enough in our experiments.
>
> If there is another approximation of KL divergence between Gaussian mixtures, that satisfies the aforementioned requirements is definitely worth to give it a try to see how the performance of our proposed algorithm changes.
>
> For sure the framework could be extended to other mixtures of distributions such as Bernoulli mixtures. The main modification will be the calculation of the new KL divergence term.
>
> I hope that we cover your questions.

---

### Decision · Program_Chairs · 2019-12-19

**Decision:**

Reject

**Comment:**

This paper proposes to use a mixture of Gaussians to variationally encode high-dimensional data through a latent space. The latent codes are constrained using the variational information bottleneck machinery.

While the paper is well-motivated and relatively well-written, it contains minimal novel ideas. The consensus in reviews and lack of rebuttal make it clear that this paper should be significantly augmented with novel material before being published to ICLR.